# PipeMEM: A Framework to Speed Up BWA-MEM in Spark with Low Overhead

**DOI:** 10.3390/genes10110886

**Published:** 2019-11-04

**Authors:** Lingqi Zhang, Cheng Liu, Shoubin Dong

**Affiliations:** Communication & Computer Network Lab of Guangdong, School of Computer Science & Engineering, South China University of Technology, Wushan Road 381, Guangzhou 51000, China; cslingqizhang@gmail.com (L.Z.); ztcattlepatato@gmail.com (C.L.)

**Keywords:** BWA-MEM, Spark, low overhead

## Abstract

(1) Background: DNA sequence alignment process is an essential step in genome analysis. BWA-MEM has been a prevalent single-node tool in genome alignment because of its high speed and accuracy. The exponentially generated genome data requiring a multi-node solution to handle large volumes of data currently remains a challenge. Spark is a ubiquitous big data platform that has been exploited to assist genome alignment in handling this challenge. Nonetheless, existing works that utilize Spark to optimize BWA-MEM suffer from higher overhead. (2) Methods: In this paper, we presented PipeMEM, a framework to accelerate BWA-MEM with lower overhead with the help of the pipe operation in Spark. We additionally proposed to use a pipeline structure and in-memory-computation to accelerate PipeMEM. (3) Results: Our experiments showed that, on paired-end alignment tasks, our framework had low overhead. In a multi-node environment, our framework, on average, was 2.27× faster compared with BWASpark (an alignment tool in Genome Analysis Toolkit (GATK)), and 2.33× faster compared with SparkBWA. (4) Conclusions: PipeMEM could accelerate BWA-MEM in the Spark environment with high performance and low overhead.

## 1. Introduction

The development of next-generation sequencing (NGS) technique generates data faster compared to the previous techniques. The increasing rate of sequencing data is even faster than the Moor Law in computer architecture. This situation raised a challenge for genome data analysis, which further put the first step of it, sequence alignment, into the breach of overwhelming data. All of these call for proper alignment methods with high accuracy that can also process such a large amount of data effectively.

Under this background, many great single-node alignment tools have been developed, for example, BWA-SW [1], BWA-MEM [2], bowtie2 [3], cushaw [4], etc. Among them, the experimental study conducted by Li [2] shows that BWA-MEM is faster and more accurate than most of the others. Because of that, BWA-MEM is a prevalent tool for genome analysis now. Numerous genome analysis pipelines deploy it for their alignment step, for example, GenomeScope [5], PhyResSe [6], SpeedSeq [7], GPCG [8], and GATK (genome analysis toolkit) [9].

Additionally, many multi-node alignment algorithms are proposed. The pattern that differently reads the alignment tasks is independent, making it well suited to be implemented in a parallel environment. There are many successful implementations of multi-node alignment algorithms, such as pBWA [10] and mpiBLAST [11], that used MPI (Message Passing Interface) to run BWA and BLAST [12] in multi-node. MerAligner [13] and S-Aligner [14] applied the HPC (High Performance Computing) technique. MerAligner was based on a successful single-node SIMD (Single Instruction Multiple Data) implementation [15] of the Smith–Waterman Algorithm [16], while S-Aligner designed a new algorithm similar to the alignment pipeline of RazerS3 [17]. parSRA [18] used OS (Operation System) APIs (Application Programming Interfaces) to run alignment tools. parSRA is a portable tool with relatively lower overhead. But all of the tools above dismissed a vital fact that big data platform is playing an increasingly critical role in the genome analysis pipeline. Alignment tools are only the beginning of a genome analysis pipeline. There would be some overhead to assemble these tools to the existing genome analysis pipeline that depends on big data technique.

The utilization of big data techniques in genome analysis resulted in the famous Genome Analysis Toolkit (GATK). GATK relied on Hadoop when it was first released. After the first release, GATK has been a widely used tool for genome analysis for a long time.

However, the map-reduce programming model in Hadoop relies on iterative access to HDFS (Hadoop Distributed File System), which drawbacks the performance improvement of applications run in the Hadoop platform. The solving of this problem gave birth to Spark. Instead of accessing HDFS, Spark processes data with a data structure, named resilient distributed dataset (RDD). Thus, Spark would bring about a higher performance for those applications that require multi-time hardware access compared with Hadoop.

GATK also exploited the power of Spark, which leads to GATK 4.0. Apart from GATK, many efforts have been done to implement genome analysis tools in the big data environment, for example, Biodoop [19], BioPig [20], Eoulsan [21], SparkSeq [22], HIVE [23], etc. Nonetheless, GATK is still the dominant one.

Under this circumstance, we believe that two tasks for the development of genome algorithm are equally important: developing a multi-node framework that supports BWA-MEM and supports the future development of a single-node alignment tool.the result of the alignment should be adaptable to the existing prevalent genome analysis tools in a big data environment, i.e., GATK.

There are some research works related to these tasks, such as BigBWA [24] and SEAL [25], that tried to run the BWA in the Hadoop environment. SparkBWA [26] is an improvement of BigBWA. Considering that BigBWA is faster than SEAL [25] and SparkBWA is faster than BigBWA [24], this research would only consider SparkBWA. Besides, BWASpark [27] is a tool in GATK4. Because GATK is a popular tool, we would also discuss the performance of BWASpark. 

BWASpark (GATK) integrates BWA-MEM inside tightly. It creates BWA threads instead of calling a whole BWA program with JNI (Java Native Interface). Before the main computation, preparations are necessary to conduct. BWASpark requires fastq files to be transformed into sam files. BWASpark also creates a self-defined index based on the index files created by BWA. These files should be loaded to HDFS firstly. In the main computation part, BWASpark would process only part of data to BWA-MEM because only sequence information is required to perform an alignment. This effort would improve its performance if the network transferring is in bottleneck condition. After alignment by BWA-MEM, the alignment result is integrated with the data extracted from the original file to generate a complete result. Results from the different tasks would then be stored in HDFS. After that, BWASpark also provides the methods for merging all separated files.

SparkBWA calls the BWA program directly. It can run all kinds of BWA tools in the Spark environment. It consists of three stages: RDD creations, map, and reduce phases. But this program requires some preparation before these main stages can be processed. For preparation, fastq files (the sequences that need to be aligned) need to be uploaded into HDFS, and the index file needs to be put to a path that can be accessible for any node. For the main stage, there are three separate stages: RDD creation step is in charge of data distribution.Mapper calls the BWA program with the help of JNI.Reducer merges the results produced by BWA in mapper together.

The reducer step can be disabled. RDD creation in SparkBWA provides two methods to handle the pair-end situation: Spark-Join and SortHDFS. Both of them use line numbers as a key-value to conduct join operation or sort operation. Spark-Join consumes a large amount of memory, but it is faster. SortHDFS is slower while saving in memory.

Now, we considered alignment as the first step of genome analysis. Loading data from local disk into HDFS was unavoidable, considering all of the following analyses were done in Spark.

Nonetheless, we believed it is unnecessary to merge the results after the main computation because the future step would also require the data to be separated and distributed again.

The current implementations of BWA in Spark have some shortcomings. BWASpark (GATK) might suffer from additional overheads of assembling and dissembling input and output data. We used a parquet, a column-oriented data storage format, to handle this problem. This storage format enhanced us to update only part of the results. We used this as part of our genome analysis pipeline, which belongs to another work. So, we would not discuss much of it in this paper, but this is not a big issue here. SparkBWA suffered from redundant works. We thought that it was possible to avoid the iteration throughout input data by adding some additional work in the preparation stage. It was also possible to prevent unnecessary disk access before calling the BWA-MEM. Besides, the usage of JNI made both projects not so easy to maintain and not so portable if users want to use another alignment tool.

The result of 61.05% and 33.64% showed that it was slower than the original C code in a single-node environment. Therefore, we proposed light and portable fame work, PipeMEM, that could avoid most of the overhead existed in counterpart products.

## 2. Methods

### 2.1. Framework Structure of PipeMEM

#### 2.1.1. Workflow of PipeMEM

Similar to BWASpark (GATK) and SparkBWA, PipeMEM consists of a pre-processing stage and the main computation stage. We would not consider merging the output files since these distributed files would be processed to the next step in genome analysis.
In the pre-processing stage: (1) Change the data format of input files so that there is no need to iterate and merge input data in the main computation step. (2) Upload data to HDFS.In the main computation stage: (1) Distribute data from HDFS to different nodes. (2) Transform the data into the original format, so that the BWA-MEM program can process it. (3) Call BWA-MEM. (4) Upload the results to HDFS.

Figure 1 shows the workflow of PipeMEM. We have to mention that this framework also works on other single-node programs that belong to map pattern parallelism, which include other alignment tools.

Table 1 shows how this framework is different from BWASpark(GATK) and SparkBWA. From this table, we could see that PipeMEM involves fewer operations compared with BWASpark. Accordingly, we could make some modifications in the data format so that we could avoid iterating fastq files in the main computation stage. Admittedly, this operation would increase the time consumption in the pre-processing stage. But our experiments showed that the benefit brought by this design outweighed its overhead.

#### 2.1.2. Data Flow of PipeMEM

As Figure 2 shows, every four-line composes one unique read, and every four-lines in two paired-end read files make up a data unit. For Spark, textFile API can separate the text files based on the line. In other words, the texFile treat line as a data unit. So, all we need to do is to design a data format to adapt the data unit concept of the fastq file to the concept of textFile.

Some identifiers are chosen that are not existed in the fastq file to replace the line breaker inside a data unit of the fastq file. In this way, after the data is distributed by Spark, it’s easy to regenerate original data, by replacing the identifier with the line breaker.

Figure 3 shows how pre-processing was developed in a paired-end situation. It uses paired-end read as an example because the paired-end is more complicated than a single end. Figure 3 also shows how the modified data format was changed back to the original format that could be processed by BWA-MEM. We used a smart pairing parameter in BWA-MEM, with which BWA-MEM would be able to read paired-end from a single file instead of two files.

Next, we have discussed two main optimization techniques: pipeline in pre-processing and in-memory-computation in main computation.

PipeMEM program is publicly available at https://github.com/SCUT-CCNL/PipeMEM.

### 2.2. Pipeline in Pre-Processing

#### 2.2.1. Design Principle

In the pre-processing, there are three tasks: (1) loading data from local disk, (2) altering data format, and (3) uploading data to HDFS. Luckily, loading and uploading belong to I/O (Input/Output) task, which can be handled by DMA (Direct Memory Access). Additionally, uploading data relies on the channel of the network, while loading data depends upon the channel of disk access, which means they are independent of each other. Moreover, modifying the data format is done by CPU. So, three different kinds of hardware can handle these three tasks separately.

According to [28], if there are a series of operations, and each operation can be mapped into a specific hardware structure, a pipeline parallel pattern is a good solution. 

We hereby proposed to use a pipeline structure. Figure 4 shows the structure of the pipeline in the pre-processing from a hardware perspective. There are three kinds of hardware structure: the first one handling loading data, the second one in charge of data format, and the third one handling the task of uploading data. Every time-unit, a set of data is processed into an independent hardware structure. 

#### 2.2.2. Implementation

In the situation of single-end, three processes are created; each process correlates to one hardware stage.

The situation of the paired-end is a little tricky. The four processes are utilized:Two of them handling loading dataOne handling data format transformationOne handling uploading data to HDFS

Though there are more processes compared with the single-end situation, 2 of the processes require the same hardware channel and corresponds to the first stage of the pipeline. This implementation would alleviate the complexity of programming, while it would not harm the performance.

Figure 5 shows how processes are mapped to different stages of pipeline in the pre-processing step when processing paired-end data.

According to Hennessy and Patterson’s work [29], if there is no overhead of pipeline, the speedup of the pipeline would be equal to the depth of the pipeline, in other words, the number of stages of the pipeline.
(1)$$Speedup=\frac{C_{stage\;count}}{1+overhead}$$
where C is the total number of stages in a pipeline or the depth of a pipeline.

In this situation, we can expect at most 3× speedup in the pre-processing.

### 2.3. In-Memory-Computation in Main Computation

#### 2.3.1. Design Principle

In the main computation stage, SparkBWA would firstly store data from HDFS to local disk. Then, it would start a BWA-MEM to process this data. Though this method is easy to implement, it results in overhead.

In order to deal with it, the data designed from HDFS would directly be feed to BWA-MEM, which is similar to BWASpark (GATK).

Moreover, the result of BWA-MEM would also directly be stored in HDFS.

#### 2.3.2. Implementation

The implementation of PipeMEM relied on standard stream and PipeRDD. It consists of several steps:Separate data from HDFS and distribute them to different nodes to create RDDsPipe the data in RDD to a program that modifies the format of input data. This step generates data format that could be utilized by BWA-MEMCall single-node BWA-MEMPipe the result of BWA-MEM to generate new RDDs, and these RDDs would then store the data in HDFS.

Figure 6 shows the data from the main computation stage of PipeMEM. After loading data from HDFS, all the data is preserved in memory throughout all the computation, until the result is uploaded to HDFS again.

This optimization would alleviate the overhead of data access to hardware disk. 

But, admittedly, BWA-MEM leveraged the pipeline structure to overlap the hardware data access with computation. So, the overhead of hardware access in BWA-MEM is not notable. Thus, the benefit might also not be remarkable.

### 2.4. Experimental Setup

#### 2.4.1. Metrics

Apart from latency, we used three different kinds of metrics for the experiments: throughput, overhead, and scale efficiency.

##### Throughput

Throughput implies the speed of a framework. Higher throughput is better. We computed throughput with the following equation:(2)$$Throughput=\frac{N_{read\;count}}{T}$$
where N is the total number of reads in a data set, and T is the total time consumption to process these reads.

##### Overhead

There is no guarantee that parallelism would always bring about better performance. Empirically, some overheads [28] might drawback the performance of a parallel program:Overhead in launching more workersOverhead in synchronizationOverhead in network communication

Now that we designed a framework to accelerate BWA-MEM in the multi-node environment, the time consumed for the BWA-MEM computation is a valid consumption, while all the other time consumption caused by different frameworks means overhead. Therefore, we proposed to measure overhead with the following equation:(3)$$Overhead=1-\frac{T_{BWA-MEM}}{T_{framework}}$$

However, because BWA-MEM is a single-node program, this metric is meaningful only in a single-node environment.

##### Scale Efficiency

Scalability is one metric that used countlessly when discussing parallel programming. But this metric sometimes does not indicate a good performance according to McSherry’s work [30].

We used the concept of scale efficiency (Hennessy and Patterson 2011), to measure scalability. We computed scalability with the following equation. Higher efficiency implies better performance.
(4)$$Efficiency=\frac{T_{1\;node}}{\left(P\times T_{P\;node}\right)}$$

#### 2.4.2. Dataset and Experimental Environment

We conducted the following experiments on a cluster with 10 nodes. The network was 1 GigE. Each node equipped with two E5-2670 CPU (2.6 GHz, 8 cores) with 64 GB memory. The Hadoop version was 2.6.4. The Spark version was 2.1.0. The BWA version was 0.7.15.

We used part of 1000 Genomes Project data (NA12750, 3.5 million read pairs, 0.98GB) and part of NA12878 data (60 million read pairs, 31 GB) to perform most of the experiments. Table 2 shows the details of the data we used.

This research has also compared PipeMEM with SparkBWA and BWASpark (GATK) in the following section. Instead of using the same setting, we used a parameter that we found could achieve the best performance in our experiment environment. We used a method similar to PipeMEM we discussed in Section 3.2.1 to discover the best parameter setting of counterpart methods. Table 3 shows the parameter setting of SparkBWA and BWASpark.

Admittedly, BWASpark (GATK)’s setting would not possibly use all the CPU resources. But our experiment showed that 16-task BWASpark was about 1.4 times faster than the 32-task version. From our experiment, 32-thread BWA-MEM could leverage the power of hyper-threading in the CPU. So, the implementation of BWA-MEM in Spark is theoretically able to make use of all the logical resources in CPU. The reason why BWASpark cannot achieve a better performance when utilizing physical CPU cores than utilizing logical CPU cores is probably that BWASpark treats Java task and C thread equally.

## 3. Results and Discussion

### 3.1. Pre-Processing

Uploading data to HDFS is unavoidable for every Spark-based alignment tool. But, considering that alignment is the first step in a whole-genome analysis pipeline, sooner or later, data should be loaded to HDFS. So, many papers dismiss this overhead [24,25,26].

In this sub-section, we proved that the pipeline pattern could accelerate the pre-processing of PipeMEM.

We only utilized the D2 dataset here since the size of the dataset would not influence the relative performance of different methods.

Table 4 shows that utilizing pipeline can achieve 2.86× speedup in the pre-processing step of PipeMEM. This result was consistent with our analysis in the previous section, where we expected the max speedup of the pipeline pattern to be threefold. We leveraged a rather simple implementation of pipeline pattern with Python code. We believe it’s possible to accelerate the program further if we tried a C implementation.

### 3.2. Main Computation

#### 3.2.1. Parameter Setting of Tasks and Threads 

Before starting the experiment of the main computation stage, it is necessary to find out the most suitable parameters for the framework.

In the situation of PipeMEM, we needed to figure out two parameters: the task count and the thread count.

Task count is related to Spark. In the main computation stage, a single task would call a single BWA-MEM program. The thread count is related to BWA-MEM. The BWA-MEM leverage threads to handle alignment tasks. These are similar to SparkBWA while different from BWASpark (GATK), which treat tasks in Spark and thread in BWA-MEM equally.

We tested every possible parameter groups of task count and thread count with dataset D1. Since every node only had 32 logical cores, thread count larger than 32 would only harm performance. So, we thought it was not necessary to test thread count larger than 32.

We also tested a situation when the task counter was larger than eight. But in these cases, the operating system sometimes starts to use swap memory, even though system memory is still sufficient enough. The use of swap memory would no doubt decrease the performance dramatically. So, we chose not to present it here.

Figure 7 shows the performance comparison of different pairs of task count and thread count. It was easy to see that four tasks with eight threads brought about the highest performance. The following experiments would be conducted under this parameter setting.

We also used this method to discover the best parameter of BWASpark (GATK) and SparkBWA. The results are presented in Table 3.

#### 3.2.2. In Memory Computation

We tested two different versions of implementations: local disk access and in-memory optimization. This experiment was under the dataset D2. Table 5 shows the result.

From Table 6, it is easy to see that in-memory optimization can speed up the main computation of PipeMEM. It is not so notable because BWA-MEM has already implemented a pipeline pattern to alleviate the disk accessing overhead of the program.

### 3.3. Comparison of PipeMEM, BWASpark(GATK), and SparkBWA

In this sub-section, we first compared the pre-processing and the main computation stage of different frameworks separately. Then, we wrapped these stages together to compare the performance of these frameworks.

#### 3.3.1. Pre-Processing 

In this sub-section, we tested the time consumption of the pre-processing of PipeMEM, BWASpark (GATK), and SparkBWA. We included all the time consumed before putting sequence data into HDFS. But we did not include the time consumption for creating index because it is static to the data set. Besides, in addition to the time consumed by BWA to create an index, BWASpark (GATK) needed another 1.7 minutes to create a self-defined index file. We did not include this either.

Table 5 shows the time consumption of the pre-processing of different methods. SparkBWA was the fastest because it used HDFS API to finish this task, which reduced overhead dramatically.

BWASpark was the slowest one because it conducted tasks serially. There were many tasks at this stage. We believed this stage could be accelerated further if it also applied a pipeline parallel pattern.

We only leveraged Python to conduct the pre-processing. But it was still comparable with SparkBWA, which utilizes native API in the Hadoop. We believed the performance of PipeMEM would be further improved if this step was implemented with C. This would be a part of our future work. 

#### 3.3.2. Main Computation

We tested the overhead of the main computation of PipeMEM. We reported the overhead of PipeMEM, BWASpark, and BWASpark in Table 7.

It was easy to see that PipeMEM was the lightest one among all three frameworks. This performance was achieved by removing all unnecessary operations in the counterpart implementations.

It’s not easy to explain why BWASpark (GATK) suffered such an overhead. BWASpark integrated BWA-MEM tightly. It was a little hard to analyze their code. We guessed this comes from the dissembling of reads and then reassembling it into the sam structure. So, this overhead might come from implementation. Additionally, BWASpark treated processes in Java and threads in C equally, which would also harm performance. This could be another overhead from implementation.

The reason why SparkBWA’s overhead was so high is that before the main computation, when processing paired-end sequencing data, it would firstly iterate throughout the input files, merge them, and then distribute data. When all of these tasks finished, the computation of BWA could be processed. Additionally, writing the data, coming from HDFS into a local disk, before it is processed to BWA-MEM, also increased the overhead. According to our knowledge of BWA-MEM, these steps are redundant. So, overhead in this framework came from unnecessary synchronization. Interestingly, this result conflicted with [26], which stated that SparkBWA could be faster than BWA in a single node. Two reasons could explain this difference: (1) They did not report the paired-end situation. (2) The CPUs they used (AMD Opteron 6262HE) do not support hyper-threading, while the CPUs we used, support this feature. According to our experiments, BWA-MEM could make good use of this feature.

Figure 8 shows the throughput of all tools run through 10 nodes. The ideal line was drawn under the assumption that BWA-MEM could be scaled with the efficiency of one.

It was easy to see that the performance of PipeMEM was the best among the three tools, and it was the closest one to the ideal line.

The throughput of BWASpark reached its highest point at seven nodes and started decreasing. This was because the speedup brought by using more nodes was less than the overhead caused by additional nodes.

The trend of SparkBWA and PipeMEM were similar because they used a similar strategy. 

It is worth mention that the distances of each point to the ideal line in the figure imply the overhead at that point. This indicates that PipeMEM had the lowest overhead among all the tools.

Figure 9 shows the scale efficiency between PipeMEM, BWASpark, and SparkBWA. It is interesting to note that SparkBWA scaled quite well in this experiment. This was consistent with the experiment done by SparkBWA [26]. However, considering the worst performance of SparkBWA in a single-node, the scalability of SparkBWA does not imply a good performance in general. It’s a normal trend that as the node increases, the efficiency of the program decreases. This decline is caused by the overhead of parallelism. The reason why the efficiency of SparkBWA increased in a two-node situation was more or less because its performance in a single-node was too bad.

The trend of the efficiency of PipeMEM and SparkBWA was similar to each other. This implies that they suffered from similar overhead for additional nodes. This is reasonable because they utilized a similar parallel strategy. 

On the other hand, the efficiency of BWASpark decreased faster. This implied its higher overhead in parallelism. We guessed this phenomenon was caused by it dissembling reads and re-assembling reads, which increase the overhead from network communication because the original data might not be stored in the node in which the computation is performed. On the other hand, PipeMEM and SparkBWA only saved results at the closest data node of HDFS (supported by the HDFS API).

To better understand the different performance between PipeMEM and BWASpark (GATK) and between PipeMEM and SparkBWA, we further reported the relative performance of PipeMEM to BWASpark (GATK) and PipeMEM to SparkBWA. Figure 10 shows the results.

From Figure 10, we could see that PipeMEM was at least 2.2 times faster than SparkBWA and at least 1.5 times faster than BWASpark. The relative performance between PipeMEM and SparkBWA was relatively stable. The relative speedup of PipeMEM to SparkBWA was increasing as the number of nodes increased. It achieved about 2.66× speedup in the 10-node environment. On average, PipeMEM was 2.33 times faster than SparkBWA and 2.27 times faster than BWASpark.

#### 3.3.3. Integrate Analysis 

To better understand the different performance of the whole process, PipeMEM, SparkBWA, and BWASpark (GATK), we took all parts, including the pre-processing and the main computation, and compared them in this section.

Because BWASpark (GATK) achieved its best performance in seven nodes, we compared the performance of different tools in a seven-node environment. The results are shown in Figure 11.

From Figure 11, it is easy to see that the pre-processing of BWASpark was too long compared with the time consumed by its main computation. Though the time consumption of the pre-processing in PipeMEM was higher compared with SparkBWA, the total performance of PipeMEM was still better. 

In order to understand the relative performance of SparkBWA and PipeMEM, we further reported the relative speedup of PipeMEM compared with SparkBWA. The result is shown in Figure 12. It was easy to see a flatter trend of this line, and the relative speedup ended at about 1.2×. This means that even considering the overhead of pre-computation, which could not be scaled, PipeMEM was still about 20% faster than SparkBWA.

## 4. Conclusions

In this paper, we presented PipeMEM, a framework to run BWA-MEM in Spark easily while having a high performance and low overhead. The framework mainly consists of two stages: pre-processing and main computation. We further designed a data format that could be used by Spark to separate data without compromising the atomicity of the original data. We optimized the pre-processing step with a pipeline parallel pattern that achieved 2.86× speedup. We also optimized the main computation stages by utilizing standard stream and PipeRDD to ensure that there is no local disk access. We tested our framework from various performance perspectives and proved that the proposed optimization could improve the performance of PipeMEM. Our solution was significantly faster than existing tools, i.e., BWASpark (GATK) and SparkBWA, with lower overhead.

## Figures and Tables

**Figure 1 genes-10-00886-f001:**
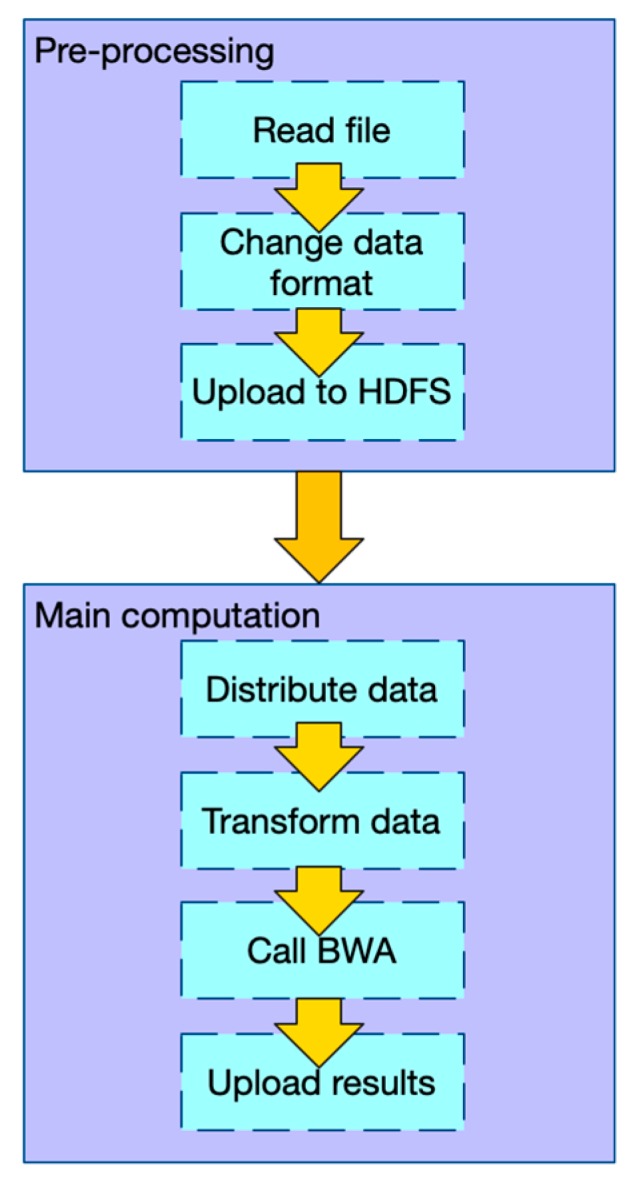
The workflow of PipeMEM in the paired-end situation. In the pre-processing stage, two input fastq files are merged and transformed, and then uploaded to HDFS (Hadoop Distributed File System). In the main computation stage, the merged file is distributed to computing nodes by Spark API (Application Programming Interfaces). In local-node, PipeMEM would redo the modification, then evoke BWA-MEM to process these data. After that, PipeMEM would upload these results to HDFS with Spark API.

**Figure 2 genes-10-00886-f002:**
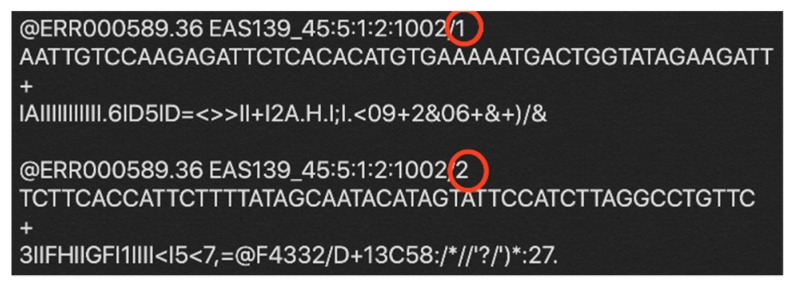
Example of paired-end reads. Four lines compose a read: (1) sequence identifier, (2) raw sequence, (3) optional line, (4) quality score. Paired-end reads share same identifiers, except the last character as showed in red circle in the figure.

**Figure 3 genes-10-00886-f003:**
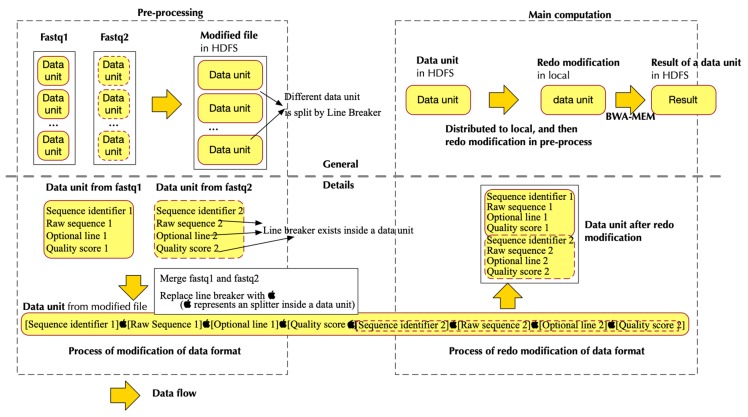
Data flow in PipeMEM in the paired-end situation. The upper part explains the process of PipeMEM with a “data unit” point of view; The lower part explains in detail how data format is modified throughout the PipeMEM process. In the pre-processing, PipeMEM would merge and transform the data unit in fastq files. Then, PipeMEM would upload the modified data unit to HDFS. In the main computation, PipeMEM would distribute the modified data unit to different nodes. PipeMEM would then transform the distributed data unit back to the original format and process them to BWA-MEM. After all of that, PipeMEM would upload the results of that data unit to HDFS again. The below part shows how data format changes in different stages of PipeMEM. In the pre-processing, PipeMEM would replace the line breaker in the data unit by a splitter. Then, PipeMEM would merge the data units in paired-end reads. In the main computation, PipeMEM would replace the splitter inside the modified data unit by line breaker again. After that, PipeMEM would feed this data unit to BWA-MEM.

**Figure 4 genes-10-00886-f004:**
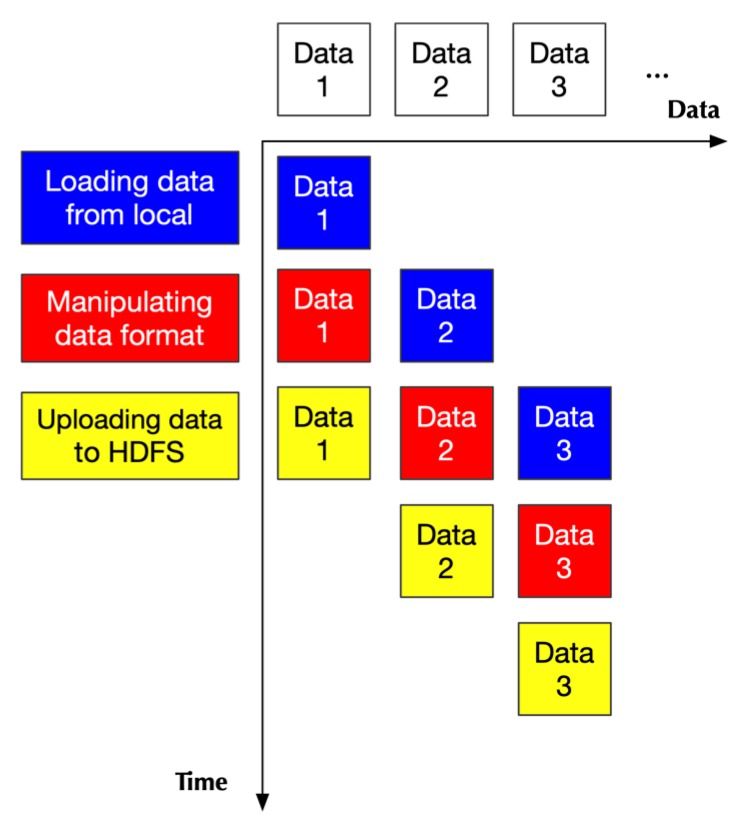
The pipeline structure in the pre-processing stage (hardware perspective). At the first time-unit, loading data hardware would process the first set of data. At the next time-unit, loading data hardware would process the next set of data, while manipulating data format hardware would start to process the first set of data. This process continues until uploading data hardware finishes the processing of the last data set.

**Figure 5 genes-10-00886-f005:**
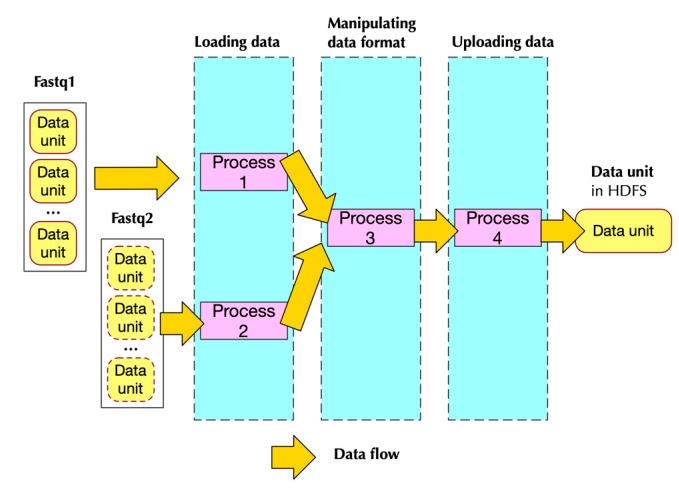
Implementation of the pre-processing stage in the paired-end situation. Process 1 and process 2 are in charge of loading data, each one handling one fastq file. They pass the input data to process 3, which is in charge of manipulating data format. Process 3 passes the result to process 4. Process 4 would upload this data to HDFS.

**Figure 6 genes-10-00886-f006:**
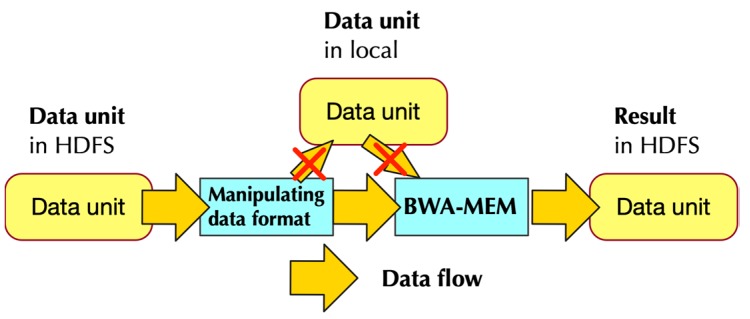
Data flow of the main computation stage. It shows that data are not stored in local-disk through the whole main computation.

**Figure 7 genes-10-00886-f007:**
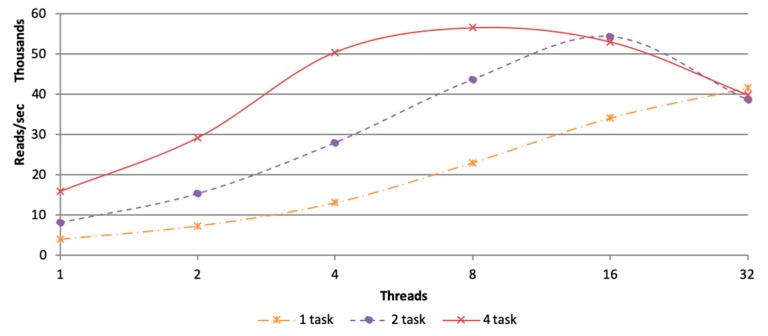
Alignment speed of different pairs of thread count and task count. Throughput of different pairs of thread count and task count, four tasks with eight threads bring about the highest performance.

**Figure 8 genes-10-00886-f008:**
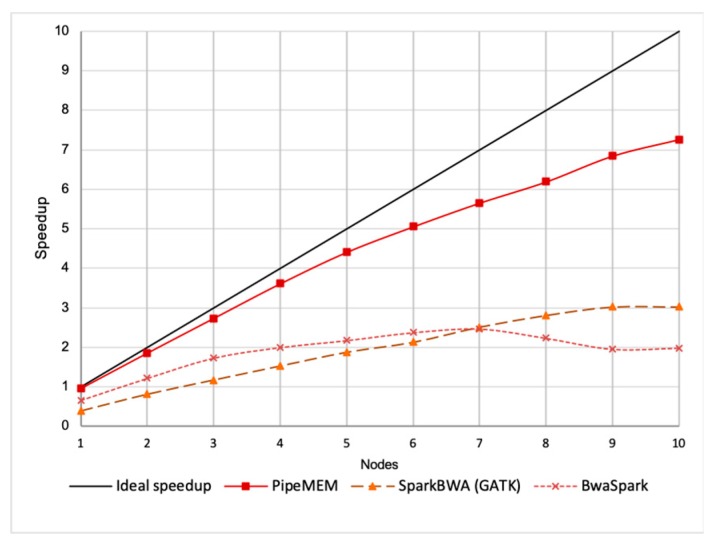
Comparison of throughput between PipeMEM, BWASpark, and SparkBWA. The ideal line is drawn under the assumption that BWA-MEM can be scaled with the efficiency of one. It’s easy to see that the performance of PipeMEM is the best among three tools, and it is the closest one to the ideal line. The throughput of BWASpark reaches its highest point at seven nodes and start decreasing. This is because the speedup brought by using more nodes is less than the overhead caused by additional nodes. The trend of SparkBWA and PipeMEM are similar because they used a similar strategy.

**Figure 9 genes-10-00886-f009:**
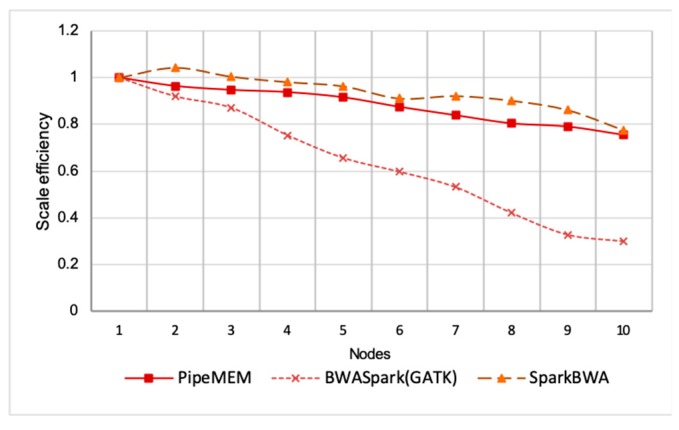
Comparison of scale efficiency between PipeMEM, BWASpark, and SparkBWA. It’s a normal trend that as the node increases, the efficiency of the program decreases. This decline is caused by the overhead of parallelism. The reason why the efficiency of SparkBWA increases in a two-node situation is more or less because its performance in single-node is too bad. The trend of the efficiency of PipeMEM and SparkBWA is similar to each other. Besides, the efficiency of BWASpark (GATK) decreases the fastest. This implies its higher overhead in parallelism.

**Figure 10 genes-10-00886-f010:**
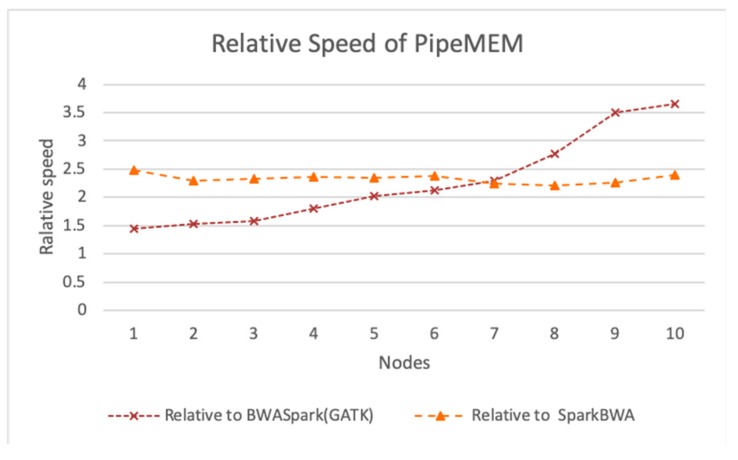
Relative speed of PipeMEM to BWASpark (GATK) and PipeMEM to SparkBWA. At every different node environment, PipeMEM maintains a higher performance compared with BWASpark and SparkBWA. PipeMEM is at least 2.2 times faster than SparkBWA and at least 1.5 times faster than BWASpark. The relative performance between PipeMEM and SparkBWA is relatively stable. The relative speedup of PipeMEM to SparkBWA is increasing as the number of nodes increases. It achieves about 2.66 × speedup at the 10-node environment. On average, PipeMEM is 2.33 times faster than SparkBWA and 2.27 times faster than BWASpark.

**Figure 11 genes-10-00886-f011:**
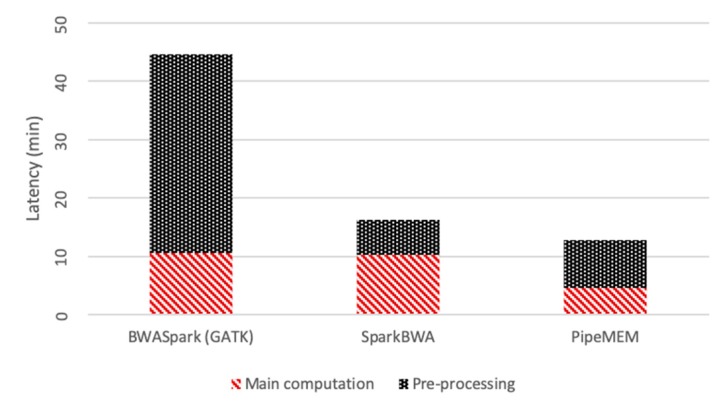
Comparison of the integrated program in seven nodes. The pre-processing of BWASpark is too long compared with the time consumed by its main computation. PipeMEM spends a relatively long time in the pre-processing compared with SparkBWA, but PipeMEM is faster, even including pre-processing.

**Figure 12 genes-10-00886-f012:**
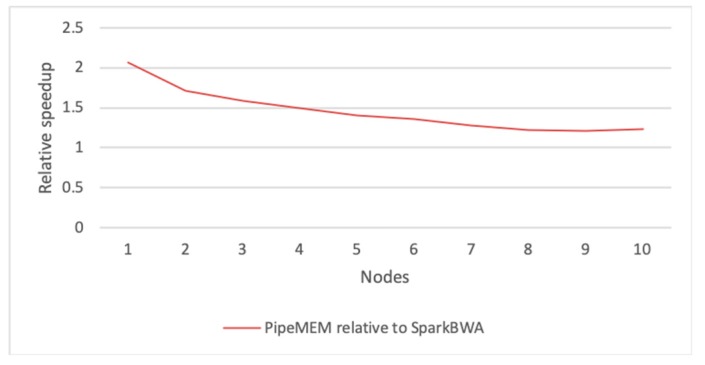
Relative speedup of PipeMEM compared with SparkBWA (including the pre-processing and the main computation). Relative speedup of PipeMEM to SparkBWA as a whole. There is a flatter trend in this line. The relative speedup ends at about 1.2×.

**Table 1 genes-10-00886-t001:** Differences between PipeMEM, BWASpark (GATK), and SparkBWA.

	PipeMEM	BWASpark (GATK)	SparkBWA
Pre-processing	(1) Change the data format (2) Upload data to HDFS	(1) Change the data format (fastq to sam) (2) Create a self-defined index file (3) Upload data to HDFS	(1) Upload data to HDFS
Main Computation	(1) Distribute data (2) Transform data back to the original format (3) BWA-MEM (4) Upload results to HDFS	(1) District sequence from the original file (2) Distribute data (sequence) (3) Modified-BWA-MEM (4) Merge results with original data (5) Store results to HDFS	(1) Iterate and then merge fastq files (2) Distribute data (3) BWA-MEM (4) Upload results to HDFS

HDFS, Hadoop Distributed File System.

**Table 2 genes-10-00886-t002:** Dataset.

Tag of Dataset	Number of Paired Read	Read Length (bp)	Size (GB)	Comment
40 pD1	350,000	51	0.98	Cut from NA12750
D2	60,000,000	100	31	Cut from NA12878

**Table 3 genes-10-00886-t003:** Parameter Setting.

Item	Task	Thread	Comment
PipeMEM	4	8	
BWASpark (GATK)	16	-	Every task corresponds to a thread
SparkBWA	2	16	Use Spark-Join instead of SortHDFS

**Table 4 genes-10-00886-t004:** Latency of pre-processing in PipeMEM.

D2	PipeMEM	PipeMEM (pipeline)	Speedup
Pre-processing (min)	23.31	8.16	2.86×

**Table 5 genes-10-00886-t005:** Time consumption of the pre-processing in PipeMEM, BWASpark, and SparkBWA.

D2	PipeMEM	BWASpark (GATK)	SparkBWA
Pre-processing (min)	8.16	34.06	5.9

**Table 6 genes-10-00886-t006:** Time consumption between whether applying disk access optimization strategy in PipeMEM.

Nodes (min)	1	2	3	4	5	6	7	8	9	10	Average
Local disk access	27.60	14.00	9.70	7.30	6.04	5.26	4.82	4.32	3.95	3.70	-
In-memory (optimize)	27.00	14.00	9.50	7.20	5.90	5.15	4.60	4.20	3.80	3.58	-
Differential	0.60	0.00	0.20	0.10	0.14	0.11	0.22	0.12	0.15	0.12	0.18

**Table 7 genes-10-00886-t007:** Overhead of PipeMEM, BWASpark, and SparkBWA.

D2	PipeMEM	BWASpark (GATK)	SparkBWA
Overhead	3.70%	33.65%	61.05%

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
