# Peer review of "PipeMEM: A Framework to Speed Up BWA-MEM in Spark with Low Overhead"

_genes, 2019, doi:10.3390/genes10110886_

Round 1

Reviewer 1 Report

This paper presents a new framework for increasing speed and reducing overhead of the alignment step in genome alignment pipelines. This framework requires more effort in data preparation but speeds up the main computation and eliminates data storage on local disks.

The graphics in the manuscript are clear, with several flow charts and a comparison table that highlights the differences in their program. It is freely available and I expect that certain users will find this useful and appealing. I am not sure that many labs would change their pipeline (involving re-training, etc.) for modest speed gains, but that can make a difference when large amounts of data are analyzed. 

I have no specific changes to suggest, other than English language style and word usage.

Author Response

Thank you for your comment and suggestion. We have proofread the manuscript again and made some modifications. And the manuscript has been revised thoroughly in language by a native English speaker.

Reviewer 2 Report

In this paper, authors proposed a framework to speed up BWA-MEM in Spark with low overhead. The sequence alignment is an essential step in genome analysis. Therefore, parallelization for sequence alignment is also very important. In this paper, authors focus on Spark framework. They carefully implement their methods and describe every step in detailed. Therefore, it is easy to read and understand. They also compare their methods with current best algorithms. It shows their methods are times faster than BWASpark and SparkBWA. Overall, their paper is worthy to read and valuable for improvement of efficiency of alignment. Yet, their paper is too lengthy and could be shorten. 

Author Response

Thank you for your comment and suggestion. After carefully proofreading, we removed some unnecessary parts.